# Secreted Signaling Molecules at the Neuromuscular Junction in Physiology and Pathology

**DOI:** 10.3390/ijms22052455

**Published:** 2021-02-28

**Authors:** Bisei Ohkawara, Mikako Ito, Kinji Ohno

**Affiliations:** Center for Neurological Diseases and Cancer, Division of Neurogenetics, Nagoya University Graduate School of Medicine, Nagoya 466-8550, Japan; ito@med.nagoya-u.ac.jp (M.I.); ohnok@med.nagoya-u.ac.jp (K.O.)

**Keywords:** neuromuscular junction, acetylcholine receptor, Wnt/β-catenin signaling, FGF signaling

## Abstract

Signal transduction at the neuromuscular junction (NMJ) is affected in many human diseases, including congenital myasthenic syndromes (CMS), myasthenia gravis, Lambert–Eaton myasthenic syndrome, Isaacs’ syndrome, Schwartz–Jampel syndrome, Fukuyama-type congenital muscular dystrophy, amyotrophic lateral sclerosis, and sarcopenia. The NMJ is a prototypic cholinergic synapse between the motor neuron and the skeletal muscle. Synaptogenesis of the NMJ has been extensively studied, which has also been extrapolated to further understand synapse formation in the central nervous system. Studies of genetically engineered mice have disclosed crucial roles of secreted molecules in the development and maintenance of the NMJ. In this review, we focus on the secreted signaling molecules which regulate the clustering of acetylcholine receptors (AChRs) at the NMJ. We first discuss the signaling pathway comprised of neural agrin and its receptors, low-density lipoprotein receptor-related protein 4 (Lrp4) and muscle-specific receptor tyrosine kinase (MuSK). This pathway drives the clustering of acetylcholine receptors (AChRs) to ensure efficient signal transduction at the NMJ. We also discuss three secreted molecules (Rspo2, Fgf18, and connective tissue growth factor (Ctgf)) that we recently identified in the Wnt/β-catenin and fibroblast growth factors (FGF) signaling pathways. The three secreted molecules facilitate the clustering of AChRs by enhancing the agrin-Lrp4-MuSK signaling pathway.

## 1. Introduction

The neuromuscular junction (NMJ) is a prototypic synapse between an ectoderm-derived element (the axon terminal of the spinal motor neuron) and a mesoderm-derived element (the skeletal muscle fiber). The NMJ uses acetylcholine (ACh) as a neurotransmitter, which is recognized by the nicotinic acetylcholine receptor (AChR) embedded in the postsynaptic membrane. Genetic defects in molecules expressed at the NMJ result in a group of neuromuscular diseases called congenital myasthenic syndromes (CMS), which are characterized by muscle weakness, accelerated muscle fatigue, muscle hypotrophy, and sometimes minor facial anomalies. In addition, signal transduction at the NMJ is compromised in a variety of diseases including myasthenia gravis, Lambert–Eaton myasthenic syndrome, Isaacs’ syndrome, Schwartz–Jampel syndrome, Fukuyama-type congenital muscular dystrophy, amyotrophic lateral sclerosis, and sarcopenia [1,2].

The NMJ in vertebrates has a typical structure. The axon terminal of a motor neuron sinks into the muscle membrane, creating a characteristic depression referred to as the primary gutter [3]. Each gutter has smaller secondary invaginations of the postsynaptic membrane. The secondary and primary invaginations are collectively called the junctional folds. AChRs are primarily clustered on the top of the postsynaptic membrane, called the synaptic crest, which directly juxtaposes the presynaptic nerve terminal. AChR clustering is mediated by an intracellular structural protein, rapsyn, which binds to the cytoplasmic domains of AChR subunits, and forms a homomeric network beneath the postsynaptic muscle membrane [4]. Rapsyn also binds to other critical muscle-derived structural molecules for the formation and maintenance of NMJ. NMJ formation is also driven by genes specifically expressed in the subsynaptic nuclei, which are located beneath the motor endplate. The subsynaptic nuclei acquire a particular chromatin organization that enables synapse-specific transcription of the NMJ molecules such as the AChR subunits [5].

The identification of CMS-associated mutations in genes specifically expressed at the NMJ contributed to the discovery of molecules involved in the NMJ signal transduction. Mutations in more than 30 genes have been identified in CMS patients. The major causes of CMS are mutations in genes encoding the AChR subunits, namely, the α1 subunit (*CHRNA1*), β1 subunit (*CHRNB1*), δ subunit (*CHRND*), and ε subunit (*CHRNE*). Other causes include mutations in genes encoding (i) structural proteins for scaffolding AChR clusters, (ii) presynaptic molecules involved in ACh release and resynthesis, (iii) intrasynaptic molecules to degrade ACh and to facilitate juxtaposition of the nerve terminal and the motor endplate, (iv) enzymes that glycosylate extracellular domains of transmembrane proteins and extracellular matrix (ECM) proteins at the NMJ, and (v) signaling molecules that drive AChR clustering [6]. In the last category (v), mutations in *AGRN*, *LRP4*, *MUSK*, and *DOK7* have been identified in CMS patients. The best-characterized synaptic signaling molecule to drive AChR clustering is the neural isoform of agrin (encoded by *AGRN*). Neural agrin is released from the nerve terminal. Two postsynaptic receptors for agrin at the motor endplate are the low-density lipoprotein receptor-related protein 4 (Lrp4; encoded by *LRP4*) and muscle-specific receptor tyrosine kinase (MuSK; encoded by *MUSK*). Dimeric Lrp4 and dimeric MuSK form a hetero-tetrameric receptor for agrin, and the receptor is associated with ECM molecules including laminin α5 (encoded by *LAMA5*), laminin β2 (encoded by *LAMB2*), collagen type XIII α1 (encoded by *COL13A*), and collagen-like tail subunit of asymmetric acetylcholinesterase (ColQ; encoded by *COLQ*), all of which belong to category (iii) above.

Identification of CMS-associated gene mutations and concomitant characterization of the mutant molecules enabled identification of physiological mechanisms behind how synaptic signaling molecules mediate AChR clustering and NMJ functioning. This review focuses on the secreted signaling molecules that facilitate AChR clustering and NMJ formation. We place special emphasis on well-characterized ones (agrin, ACh, Wnt, and neuregulin-1), and newly identified ones (Ctgf, Rspo2, and Fgf18) (Figure 1).

## 2. Environment of the Signaling Molecules: Extracellular Matrix (ECM)

The synaptic structure of the NMJ depends on several molecules. Transmembrane molecules at the NMJ form a large protein complex together with ECM molecules, intracellular molecules (e.g., dystrophin, rapsyn, and utrophin), and downstream signaling molecules to stabilize AChR clusters at the NMJ. For example, the extracellular domain of α/β-dystroglycans anchors an ECM protein, perlecan. Perlecan binds to another ECM protein, ColQ, and ColQ binds to transmembrane protein, MuSK. The sarcoglycan complex comprises the transmembrane proteins and the α, β, δ, and γ subunits; the α subunit binds to another transmembrane protein, Lrp4 [7,8,9]. α/β-Dystroglycans and the sarcoglycan complex form a dystrophin-glycoprotein complex (DGC), which also includes cytoskeletal scaffolding molecules dystrophin and utrophin that link the cytoskeleton to the NMJ [10].

Representative synaptic ECM molecules are collagen fibrils, nidogen 2, laminins (e.g., laminin subunits α2/α5/β2, encoded by *LAMA2*, *LAMA5*, and *LAMB2*, respectively), and proteoglycans (e.g., perlecan and biglycan). Six collagens (IV, VI, XIII, XXV, XXVIII, and Q) are enriched at the NMJ, and two of them (XIII and Q) are defective in CMS [11]. Collagen fibrils bind to other ECM molecules, including nidogen 2, laminins, integrins, and perlecan. Laminins are hetero-trimers that critically maintain both presynaptic and postsynaptic NMJ structures by binding to transmembrane proteins, including integrins, dystroglycans, and voltage-gated calcium channels [12]. Laminins, nidogen 2 [13], and collagen IV/XIII also bind to other ECM molecules to form a collagen-laminin network at the NMJ. Perlecan concentrated at the NMJ is an acceptor molecule for ColQ-tailed acetylcholinesterase (AChE encoded by *ACHE*) at the NMJs [14]. The heparin/heparan sulfate chains of perlecan anchor growth factors include Wnts, fibroblast growth factors (FGFs), and bone morphogenetic proteins (BMPs) [15]. Two forms of biglycan are expressed in the skeletal muscle, whose amino acid sequences do not differ. However, one form bears chondroitin or dermatan sulfate side chains, whereas the other lacks these side chains. Biglycan binds to MuSK and α/β-dystroglycans [16].

To summarize, junctional ECM molecules constitute a critical scaffold that is responsible for conveying essential signals for the formation of the NMJ [9].

## 3. Agrin-Lrp4-MuSK Signaling Pathway

Missense mutations in *AGRN*, *LRP4*, and *MUSK* cause CMS. The agrin-Lrp4-MuSK signaling pathway is the primary mechanism for the formation of NMJs. The agrin ligand is secreted from the nerve terminal of a spinal motor neuron. Its receptors, Lrp4 and MuSK, are located on the postsynaptic muscle membrane. Since the first identification of a complex comprising agrin, Lrp4, and MuSK in 2008 [17,18], its structural basis has been resolved [19]. On the postsynaptic membrane of the NMJ, dimeric Lrp4 binds to dimeric MuSK [18,20]. Two molecules of agrin and two molecules of Lrp4 form a hetero-tetramer, presumably as an asymmetric tetrameric unit [20,21]. The binding of agrin to Lrp4 is a key step in MuSK activation and in the initiation of downstream signaling. Co-immunoprecipitation studies have revealed that agrin strengthens the binding of the ectodomains of Lrp4 and MuSK in an allosteric manner [22]. We also reported that agrin enhanced the binding of Lrp4 and MuSK approximately 36-fold, as found using an in vitro plate-binding assay [23]. Thus, an asymmetric hetero-hexamer comprising two agrin molecules, two Lrp4 molecules, and two MuSK molecules, activates MuSK to induce AChR clustering.

Phosphorylation of MuSK is triggered by agrin and is further enhanced by a cytoplasmic adapter protein, downstream of kinase 7 (Dok7) [24]. Furthermore, Dok7 forms a dimer to activate MuSK by recruiting other adapter proteins, including the CT10 regulators of kinase (Crk and Crk-L) [25] and Sorbin and SH3 domain (Sorbs) 1/2 [26], which are required for stabilizing AChR clusters at the NMJ. Downstream signaling molecules of Crk, Crk-L, and Sorbs include Rho GTPases, disheveled (Dvl), p21-activated kinases (PAKs), tumorous imaginal disc 1 (TID1), Abl1/2, cortactin, actinin, microtubule-actin crosslinking factor 1 (MACF1), actin, and myosin [27]. However, the exact pathways leading to AChR clustering remain to be fully defined.

Targeted disruption of *Agrn* in mouse spinal motor neurons, as well as of *Lrp4* and *Musk* in the mouse skeletal muscle, has been reported. Homozygous agrin-deficient mice develop normally until the last embryonic day (E18.5) but die around birth due to respiratory failure [28,29]. Furthermore, skeletal muscle-specific knockouts of *Lrp4* [30] and *Musk* [31] show no AChR clustering at the NMJ at E18.5, and the mice die immediately after birth. CMS and genetically engineered mice suggest that agrin-Lrp4-MuSK signaling is the most important pathway for AChR clustering (Figure 1), and that this receptor complex is a potential therapeutic target for neuromuscular diseases [1], as well as for aging [32].

### 3.1. Agrin

#### Agrin-Binding Partners: Laminins and α/β-dystroglycans

The N-terminus of agrin binds to the ECM network via laminins α2/α5/β2. Agrin is also anchored to α/β-dystroglycans at the NMJ. By binding to these ECM molecules, agrin contributes to stabilize the synapse and maintain its structural integrity via connection to the ECM network [33,34,35]. 

Domains: More than 10 *AGRN* variants have been reported in severe forms of CMS and some of them have been partially characterized to determine their effect on the NMJ [36,37,38,39,40,41,42,43]. Agrin domains and CMS-causing mutations are summarized in Figure 2. The N-terminal agrin (NtA) domain is important for the formation of synaptic ECM, because it binds to laminins [33,36]. The p.G76S and p.N105I variants identified in the laminin-binding domain in CMS hinder the formation of AChR clusters in C2C12 myotubes [36].

The sperm protein, enterokinase, and agrin (SEA) domain is important for the glycosylation and secretion of agrin, and also protects agrin from degradation by proteases such as neurotrypsin [41]. Indeed, p.L1176P [41] and p.S1180L [44] in the SEA domain cause the instability of the agrin protein and impair the formation of AChR clusters in C2C12 myotubes. Between the NtA and SEA domains, nine follistatin (FS), two laminin EGF-like (LE), and one serine/threonine-rich (S/T) domains exist. Although the functions of the FS, LE, and S/T domains remain elusive, they contain glycosylated sites [45], and are thus thought to be important for binding to neural cell adhesion molecule 1 (NCAM1) through the heparan sulfate chain [46]. 

The first and second laminin G-like domains (LG1 and LG2) are essential for anchoring agrin to the NMJ using α-dystroglycan or other muscle-specific cell surface molecule(s) [44,47]. Two mutations (p.P1448L and p.G1509W) in the LG1 domain lead to CMS [43,44]. Although p.P1448L remains to be functionally characterized, p.G1509W in LG1 affects the binding of agrin to sulfated glycosaminoglycans, which are essential for the accumulation of agrin at the NMJ [44]. 

The LG2 domain includes codons encoded by the neuron-specific alternative exon A/y [38,48] and binds to heparin [49]. Five mutations in the LG2 domain (p.G1675, p.R1698C, p.G1709R, p.V1727F, and p.A1768P) cause CMS [39,41,42,44,50]. p.G1675S reduces AChR clustering in C2C12 myotubes, but has no effect on MuSK phosphorylation in C2C12 myotubes or in HEK293 cells [44]. The p.R1698C variant induces the instability of mutant agrin and reduces AChR clustering in C2C12 myotubes [41]. p.G1709A destabilizes endogenous NMJs when the mutant agrin is injected into rat soleus muscle [37]. p.V1727F markedly attenuates MuSK phosphorylation and AChR clustering in C2C12 myotubes [38,42]. p.A1768P was reported in a family of CMS patients without functional characterizations [39]. 

The third LG (LG3) domain contains codons encoded by two neuron-specific alternative exons named B/z, and is essential for inducing AChR clusters by binding to Lrp4 and thus activating MuSK [37]. p.G1871R in LG3 was reported in a CMS patient without functional characterizations [36]. p.Y1877D in LG3 compromises AChR clustering and MuSK phosphorylation in C2C12 myotubes [44]. p.Y1877D is located at seven codons upstream to the alternatively inserted B/z codons, which confers Lrp4 binding [20]. Thus, the analysis of p.Y1877D corroborated the hypothesis that the LG3 domain with B/z exons is essential for inducing AChR clusters by binding to Lrp4 and activating MuSK.

### 3.2. Low-Density Lipoprotein Receptor-Related Protein 4 (Lrp4)

#### Lrp4-Binding Partners: Amyloid β Precursor Protein (App), α-Sarcoglycan, and Connective Tissue Growth Factor (Ctgf)

In addition to agrin and MuSK, Lrp4 binds to App [51,52], App-like protein 2 (Aplp2) [51,52], α-sarcoglycan [8], and Ctgf [53]. App and Aplp2 are membrane proteins expressed in both the nerve terminal and the postsynaptic muscle membrane, and contribute to agrin-Lrp4-MuSK signaling–mediated presynaptic specialization [51,52]. α-Sarcoglycan is a constituent of a sarcoglycan complex, which binds to α/β-dystroglycans and forms DGC to link to the cytoskeleton at the NMJs [10]. In addition, α-sarcoglycan directly binds to the polypeptide core of biglycan, which interacts with collagen VI, to link to the ECM to the NMJs [54]. We recently reported that Ctgf also binds to Lrp4 and enhances Lrp4 binding to MuSK at the NMJ [55]. Ctgf is an ECM-associated protein that binds to integrin receptors, heparin sulfate proteoglycans, perlecan, and slit guidance ligands. Additionally, Lrp4 binds to extracellular components of the Wnt signaling pathway, including Wnt ligands, dickkopf [56], sclerostin [57], gremlin [57], wise [57], and apoE [58].

Domains: Lrp4 is a transmembrane molecule that contains extracellular and intracellular domains (Figure 2). The intracellular domain at the C-terminus contains an NPxY motif and a PDZ-interacting motif. The extracellular domains include a signal peptide, eight low-density lipoprotein receptor domain class A (LDLa) repeats, five intervened EGF-like domains, and four YWTD domains. The YWTD domain of Lrp4 forms a β-propeller structure that is observed in low-density lipoprotein receptors [59]. LDLa repeats 6–8, the first β-propeller domain, and the two intervening EGF-like domains in Lrp4 are sufficient to bind to neural agrin [21]. Similarly, the C-terminal moiety of LDLa repeats and the third β-propeller domain of Lrp4 is critical for MuSK binding [21]. Mutations in *LRP4* cause not only CMS but also (i) Cenani–Lenz syndactyly syndrome (CLSS), which is characterized by syndactyly; (ii) sclerosteosis 2 (SOST2), which is characterized by sclerosing bone dysplasia; (iii) Richter syndrome (RS), which is characterized by the transformation of chronic lymphocytic leukemia to aggressive lymphoma; and (iv) a predisposition to low-trauma fracture (LTF) [60]. CLSS is caused by mutations in p.D137N and p.C160Y in the LDLa repeats; p.D449N, p.T461P, p.L473F, and p.D529N in the first β-propeller domain; and p.L953P and p.C1017R in or C-terminal to the second β-propeller domain [61]. In the third β-propeller domain, p.R1170W and p.W1186S were reported in patients with SOST2 [62], while p.E1233K, p.E1233A, and p.R1277H occurred in patients with CMS [59,63]. We showed that the SOST2 mutations were located in the central cavity of the third β-propeller domain and compromised binding of Wnt ligands, whereas the CMS mutations were at the edge of the third β-propeller domain and compromised binding of agrin [59,63]. Thus, the third β-propeller domain of Lrp4 functions in the Wnt and agrin signaling pathways in a position-specific manner. Single-nucleotide variants in the third β-propeller domain (p.I1086V and p.A1203V), in the fourth β-propeller domain (p.S1554G), and C-terminal to the fourth β-propeller domain (p.R1646Q) are also associated with an increased risk for LTF due to decreased bone mineral density and RS [64,65,66]. The extracellular region of Lrp4, which comprises 90% of the amino acids of Lrp4, is sufficient to induce AChR clustering at the NMJs [22], and the function of the cytoplasmic region of Lrp4 remains to be determined. Lrp4 also transduces retrograde signals to modulate presynaptic differentiation, independent of MuSK activation [30,67].

### 3.3. Muscle-Specific Receptor Tyrosine Kinase (MuSK)

#### MuSK-Binding Partners: Collagen Q (ColQ), Biglycan, and Neuregulin-1/Erb2/4/Erbin

In addition to Lrp4, the extracellular domain of MuSK directly binds to ColQ and biglycan [23]. A triple helical ColQ binds to 12 AChE molecules to form an asymmetric form of AChE. Thus, a complex comprising MuSK, ColQ, and biglycan localizes AChE at the NMJ, and mediates AChR clustering and the stabilization of the postsynaptic structure. A recent study indicated that MuSK also binds to another secreted signaling molecule, BMP4 [68]. Multiple intracellular molecules that interact with the intracellular domain of MuSK were previously reviewed [1], and are not addressed in detail here. 

Domains: The extracellular domain of MuSK has four immunoglobulin-like domains (Ig1–4) (Figure 2). Ig1 and Ig2 domains mediate agrin-Lrp4 signaling [69] by binding to Lrp4 for forming a hetero-tetramer of MuSK and Lrp4 [20,21]. The cysteine-rich domain (CRD) of MuSK participates in the dimerization of MuSK on the cell surface [19] as well as in the autophosphorylation of the intracellular kinase domains of MuSK via Dok7 [70]. Most *MUSK* mutations are clustered in the intracellular tyrosine kinase domain, whereas three missense mutations have been reported in CMS patients without functional characterizations: p.N103S [71,72] and p.D38E [73] in the Ig1 domain, and p.P344R [74] in the Frizzled-like cysteine-rich domain (Fz-CRD) domain. The genotype-phenotype correlations of 27 *MUSK* mutations that cause CMS have been reviewed by others [75]. Additionally, binding partners of the intracellular domains of MuSK have been reviewed in detail elsewhere [76], and will not be addressed in this review. 

## 4. Acetylcholine (ACh)

Although ACh is a neurotransmitter for the NMJ synapse, it also modulates postsynaptic AChR clustering via regulation of extrajunctional gene expression, as has been recently reviewed elsewhere [5]. When an action potential arrives at the nerve terminal, synaptic vesicles containing ACh are fused to the membrane using a SNARE complex. ACh is then exocytosed into the NMJ space, and binding of ACh to the postsynaptic AChR opens the cationic channel, allowing ion current through the postsynaptic membrane. This results in a local depolarization that initiates a muscle action potential, and triggers excitation/contraction coupling in the muscle fiber. Action potentials in the muscle strongly affect gene expression there, mainly by repressing the expression of myogenic regulatory factors (MRFs; MyoD, Myf5, myogenin, and MRF4), leading to the loss of expression of the embryonic AChR γ subunit (*Chrng*), as well as AChR α1 subunit (*Chrna1*) and *Musk*, in the extrajunctional region. The changes in gene expressions cause decreased stability of AChR in the plasma membrane. Muscle contractions suppress the expression of extrajunctional AChR subunits by increasing calcium flux [77,78], which subsequently activates protein kinase C (PKC) and Akt kinases, which then induce the expression of transcriptional repressors of dachshund family transcription factor 2 (Dach2), as well as histone deacetylases 4 and 9 (HDAC4 and HDAC9, respectively) [79,80,81,82,83]. Altogether, ACh released at the NMJ leads to the expression of adult AChR (composed of α1, β1, δ, and ε subunits) in the subsynaptic nuclei and loss of expression of embryonic AChR (γ instead of ε subunit) in the extrajunctional nuclei.

## 5. Wnt Signaling Pathways

Wnt ligands can activate three pathways: (i) the canonical Wnt/β-catenin signaling pathway, (ii) the non-canonical planar cell polarity pathway, and (iii) the non-canonical Wnt/Ca^2+^ pathway [84]. The Wnt signaling pathway and agrin-Lrp4-MuSK signaling pathway engage in crosstalk. In the cytoplasm, Dvl1, a central effector of all the canonical and non-canonical Wnt signaling pathways, interacts with MuSK to regulate AChR clustering [85,86]. The presence of the adenomatous polyposis coli (APC) and β-catenin at the NMJ indicates that the other components of the canonical Wnt pathway might also be enriched at the NMJ [87,88]. In muscle-specific β-catenin knockout mice, the presynaptic deficits in NMJ are rescued by the transcriptional domain of β-catenin, but not by the cell adhesion domain of β-catenin. As muscle-specific knockout of β-catenin causes major defects in the presynaptic nerve terminal, β-catenin is proposed to regulate a retrograde signal [89]. β-catenin induces the expression of Slit2, which is required for synaptophysin puncta in the axon terminal of the spinal motor neuron (SMN) [89,90]. NMJ defects caused by the deletion of the Wnt-binding domain of MuSK are compensated for by treatment of lithium chloride, an inhibitor of GSK3, which activates the canonical Wnt pathway by preventing β-catenin degradation [91].

In addition to the cytoplasmic crosstalk between the agrin and Wnt signaling pathways stated above, these pathways also engage in crosstalk at the extracellular level of the NMJ. A representative example at the extracellular level is observed in Lrp4 [92]. Lrp4 inhibits in the Wnt signaling pathway via binding to extracellular molecules including Wnt ligands [56], Dkk1 [56], sclerostin [57], gremlin [57], wise [57], and apoE [58]. As stated in the earlier section on Lrp4, mutations in *LRP4* cause not only CMS but also CLSS, SOST2, RS, and low-trauma fracture (LTF) [60]. Interestingly, in mice, the pre-patterning of AChR clusters requires Lrp4, but not the frizzled-like domain of MuSK—a binding domain for muscle-derived Wnt [93]. In contrast, in zebrafish, the pre-patterning of AChR clusters *does* require the frizzled-like domain of MuSK but does not require Lrp4. The difference between mice and zebrafish is likely due to an additional Kringle domain in the MuSK extracellular region in zebrafish. 

Each Wnt ligand has a unique role in AChR clustering [94]. Wnt4, Wnt9a, Wnt9b, Wnt10b, Wnt11, and Wnt16 enhance AChR clustering even in the absence of agrin, whereas Wnt3a, Wnt7a, and Wnt8a antagonize agrin-induced AChR clustering. The redundancy of Wnt ligands and their frizzled receptors makes it difficult to extrapolate the results from cultured cells to the NMJ in vivo. In addition, secreted antagonists such as secreted frizzled-related protein (Sfrp) and secreted agonists such as R-spondins (Rspos) in the Wnt signaling pathway enable the fine spatiotemporal tuning of NMJ formation. The Wnt antagonist Sfrp1 is induced by denervation at the rat NMJ and suppresses Wnt signaling to disperse AChR clusters. We reported that R-spondin 2 (Rspo2) is an agrin-independent enhancer of MuSK phosphorylation and AChR clustering at the NMJ [95]. Rspo2 binds to leucine-rich repeat–containing G-protein, Lgr5, on the motor endplate, and phosphorylates MuSK to induce AChR clustering in an agrin-independent manner. We also showed that spinal motor neuron–derived Rspo2 plays a major role in AChR clustering and postsynaptic NMJ formation, and muscle-derived Rspo2 also plays a substantial role in the juxtaposition of the active zones and synaptic folds [96].

## 6. Fibroblast Growth Factors (FGFs) and Their Receptors

Muscle-derived factors regulate synaptogenesis by promoting differentiation of the nerve terminal at the NMJ. The assembly of the NMJ is initiated when the nerve and the muscle first contact each other via filopodial processes of the nerve terminal. These filopodial processes enable a close interaction between the synaptic partners, which facilitates synaptogenesis. Embryonic spinal neurons of *Xenopus* preferentially extend filopodia toward co-cultured muscle cells [97]. This filopodial movement is induced by basic fibroblast growth factor (bFgf) produced by muscle, which activates neuronal FGF receptor 1 (Fgfr1) at the nerve terminal during synaptogenesis [97]. While multiple FGF ligands and multiple FGF receptors are expressed at the NMJ, little is known about their roles at the NMJ. Muscle-derived Fgfs of the 7/10/22 subfamily promote clustering of synaptic vesicles at the nerve terminal [98]. Presynaptic Fgf receptor 2 (Fgfr2) is involved in the formation of the NMJ, but its ligand remains to be identified [98]. We reported that Fgf18-knockout homozygous (*Fgf18*-/-) mice showed an abnormal aggregation of multiple nerve terminals in the diaphragm. In particular, the nerve terminals made a giant pre-synapse, but had few synaptic vesicles within each terminal [99]. Thus, Fgf18 is likely to be a specific ligand that activates presynaptic Fgfr2. In addition to the effects of Fgf18 on the nerve terminal, we observed that *Fgf18*-/- mice showed simplified motor endplates and reduced gene expression of NMJ-specific *Chrne* and *Colq* in the diaphragm [99].

## 7. Neuregulins/ErbB Signaling Pathway

Four types of neuregulins and their receptors, ErbB receptors (ErbB1-4), are expressed in motor neurons, muscle cells, and Schwann cells [100]. Neuregulin-1 (Nrg1) is involved in the expression of NMJ-specific genes, such as *Chrne* [101], in the subsynaptic nuclei [5]. Homozygous *Nrg1*-knockout mice showed a 50% reduction in the density of postsynaptic AChR; however, neither homozygous Nrg1-, ErbB2-, nor ErbB4-knockout mice showed embryonic or perinatal lethality [102,103,104]. The interaction of Nrg1 with ErbB receptors contributes to MuSK phosphorylation through an intracellular molecule, erbin [105]. The interaction of Nrg1 with ErbBs stabilizes the postsynaptic apparatus through phosphorylation of α-dystrobrevin, thus contributing to anchoring AChR clusters on the postsynaptic membrane [106]. 

## 8. Other Signaling Pathways for AChR Clustering

Hippo signaling pathway: Yes-associated protein 1 (Yap1) is a major intracellular effector of the Hippo pathway that regulates cell proliferation and differentiation during development. Activation of the Hippo/Yap1 signaling pathway restricts tissue growth in adult animals. Skeletal muscle-specific knockout of *Yap1* in mice exhibits remarkable presynaptic deficits, as well as abnormal AChR clustering. *Yap1*-deficient mice show reduced protein levels of β-catenin in skeletal muscle, suggesting that Yap1 promotes synaptogenesis through β-catenin [107]. However, endogenous ligands and receptors of the Hippo/Yap1 signaling pathway to induce AChR clustering remain elusive.

Target of rapamycin (TOR) signaling pathway: TOR proteins mediate a key signaling pathway that is responsible for muscle metabolism. Namely, TOR activity is essential for muscle development and growth; however, overactive TOR signaling is also implicated in aging and sarcopenia [108]. Mice deficient for a component of the TOR signaling pathway, the mechanistic TOR complex 1 (TORC1 encoded by *CRTC1*), show fragmented AChR clustering at 12 months of age. The roles of TOR in the development and maintenance of the NMJ were recently reviewed in detail [109]. Still, it remains unknown whether endogenous growth factors in the TOR signaling pathway, such as insulin, regulate AChR clustering.

NT-4/TrkB signaling pathway: TrkB kinase activity maintains synaptic function and structural integrity of adult NMJs. Inhibition of TrkB kinase activity or a lack of NT-4 results in reduced complexity of AChR clustering in mice [110,111].

ATP/P2X2 signaling pathway: Extracellular ATP can affect the development of NMJs with its purinoreceptors P2X2 and P2Y [112,113]. In differentiated myotubes, activation of the purinergic receptor P2Y2 induces *Ache* expression [114]. 

EphA/Ephexin1 signaling pathway: Ephexin1 is an intracellular protein linking Eph receptors and RhoA signaling in muscle cells [115]. Ephexin1 is required for the topological transformation of AChR clusters through RhoA activation [116]. Ephrins are ligands for Eph receptors, but their roles in AChR clustering remain to be defined.

BMP signaling pathway: The growth, assembly, and maintenance of the NMJ are regulated by retrograde signaling of Decapentaplegic, a BMP homolog, in *Drosophila* [117]. In mammals, MuSK is a co-receptor for BMP ligands in myotubes [68] and binding of BMP4 to MuSK induces gene expression of *Dok7*, *Wnt11,* and *Musk* in this cell type. However, the effects of BMP ligands on AChR clustering at the NMJ remain to be studied.

## 9. Therapeutic Perspectives

The NMJ is a prototypic synapse, and the molecules and mechanisms that drive the formation and maintenance of the NMJ are likely to be similar to those of synapses in the central nervous system (CNS). An understanding of the mechanisms of formation and stabilization of the NMJ will thereby propagate our understanding of the less accessible synapses in the CNS.

Therapeutic intervention at the level of protein-anchoring is possible in human neuromuscular diseases and during aging [118]. An ECM protein carries the proprietary domain(s) for anchoring to its partner molecule(s). For example, ColQ binds to MuSK via its C-terminal domain and to perlecan via its heparan sulfate proteoglycan-binding domain [119]. ColQ-tailed AChE that was expressed throughout the skeletal muscle fibers was efficiently and specifically anchored to the NMJ, and rescued the lethal phenotype of *Colq*-deficient mice to the levels of wild-type mice [120]. We named our strategy protein-anchoring therapy [9]. We also showed that protein-anchoring therapy of biglycan ameliorated a mouse model of Duchenne muscular dystrophy [121]. The protein-anchoring strategy is likely to be applicable to a broad spectrum of human diseases that are caused by deficiency of an ECM molecule.

Further insights into the mechanisms of extracellular signaling molecules at the NMJ are expected to lead to the development of therapeutic approaches for alleviating human neuromuscular diseases.

## Figures and Tables

**Figure 1 ijms-22-02455-f001:**
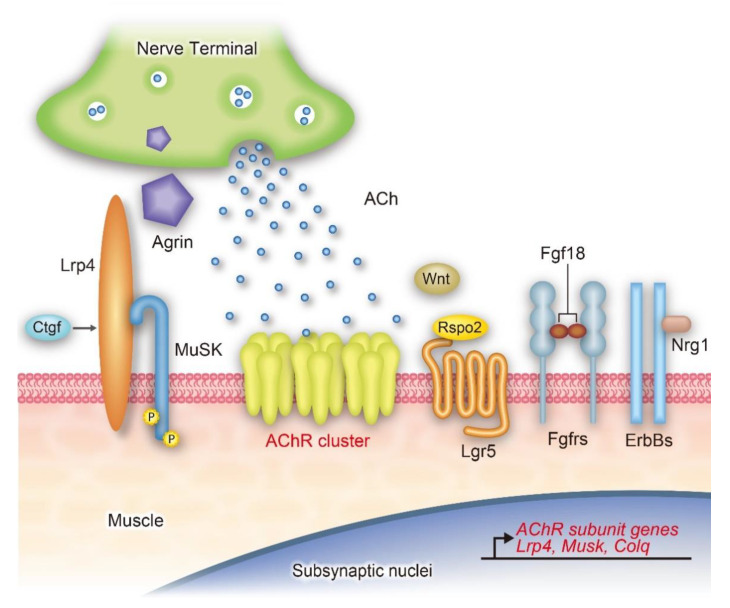
Schematic of the neuromuscular junction and representative signaling molecules involved in nicotinic acetylcholine receptor (AChR) clustering by muscle-specific receptor tyrosine kinase (MuSK) phosphorylation, and/or by gene expression. Agrin binds to lipoprotein receptor-related protein 4 (Lrp4) to phosphorylate MuSK (Section 3). Connective tissue growth factor (Ctgf) also binds to Lrp4 to enhance MuSK phosphorylation (Section 3). Wnt modulates MuSK phosphorylation by associating with either Frizzled (not shown) or Lrp4, or both (Section 5). Rspo2 binds to Lgr5 to enhance MuSK phosphorylation (Section 5). Fibroblast growth factor 18 (Fgf18) binds to Fgfrs (Section 6), and neuregulin-1 (Nrg1) binds to ErbBs (Section 7), to enhance MuSK phosphorylation and induce expression of neuromuscular junction (NMJ)-specific genes.

**Figure 2 ijms-22-02455-f002:**
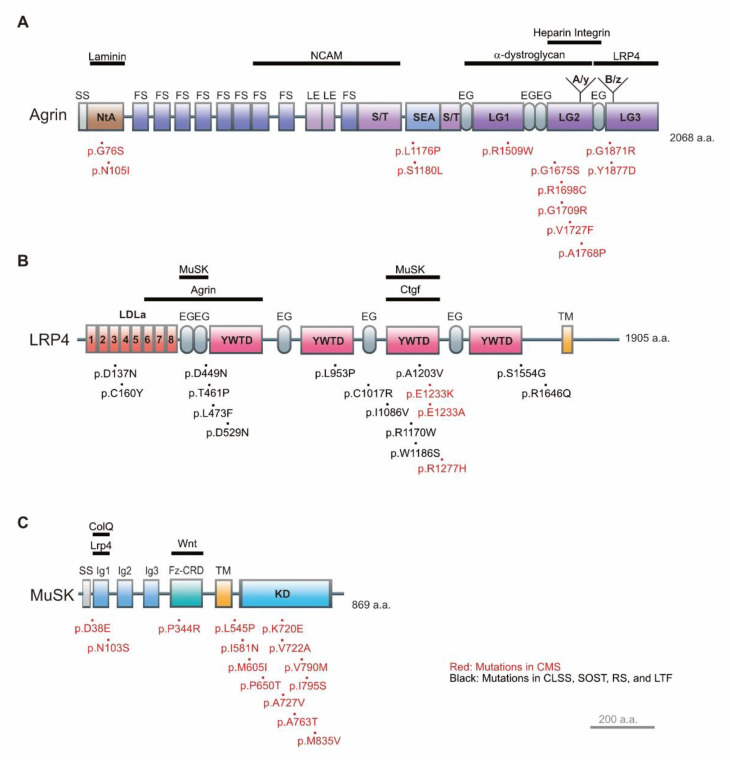
Structures and disease-causing missense mutations of agrin, Lrp4, and MuSK. The domains of agrin, Lrp4, and MuSK are indicated by squares and ovals, and are drawn to scale. Bars on the top indicate the positions where binding to a partner occurs. Mutations in congenital myasthenic syndrome (CMS) are indicated in red, and mutations in Cenani–Lenz syndactyly syndrome (CLSS), sclerosteosis type 2-like symptoms (SOST2), Richter syndrome (RS), and low-trauma fracture (LTF) are indicated in black. (**A**) Human agrin domains. SS, secretion signal peptide; NtA, N-terminal Agrin; FS, follistatin-like domain; LE, laminin EGF–like domain; S/T, serine/threonine-rich domain; SEA, a sperm protein, enterokinase, and agrin domain; EG, EGF-like domain; and LG1-3, laminin G-like domain 1–3. A/y and B/z are alternative exons included in the neuronal isoforms of agrin. (**B**) Human Lrp4 domains. LDLa, low-density lipoprotein receptor domain class A; EG, EGF-like domain; YWTD, β-propeller domain; TM, transmembrane domain. (**C**) Human MuSK domains. SS, secretion signal peptide; Ig1-3, immunoglobulin domain 1–3; Fz-CRD, Frizzled-like cysteine-rich domain; TM, transmembrane domain; and KD, kinase domain.

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
