# Peer review of "Secreted Signaling Molecules at the Neuromuscular Junction in Physiology and Pathology"

_ijms, 2021, doi:10.3390/ijms22052455_

Round 1

Reviewer 1 Report

The manuscript IJMN 1108373, entitled Secreted signaling molecules at the neuromuscular junction in physiology and pathology from B. Ohkawara et al., reviews the role of secreted molecules (agrin, etc.) during NMJ development and stabilization by changing the embryonic ACh receptor by the adult form one, and their relationships with human diseases in which NMJ is affected. Overall, the paper is well written, and their content is very interesting. However, there are some concerns that impede that the paper will be suitable for publication in present form.

(i) Lines 28-30. The authors say, “The neuromuscular junction (NMJ) is a synapse between the spinal motor neuron and the skeletal muscle fiber, and is a prototypic neuronal synapse of the central nervous system (CNS).” This definition is wrong; the neuromuscular junction is a synapse of the Peripheral Nervous System in which an ectoderm derived element (the axon terminal) contacts with aa mesoderm derived element, the muscle fiber (see Peters, Palay and Webster (1976) The Fine Structure of the Nervous System: The Neurons and Supporting Cells. W.B. Saunders Co., PA, pp. 120-124).

(ii) Figure 1. Although mutations eliciting CMS are stated either in red or black, and their description is clear through the text of the paper (lines 166-277). The inclusion of the acronym of each CMS near to the mutation (red or black) in the figure will help to the reader.

Author Response

Responses to reviewer 1

We appreciate productive comments.

Comment (i) Lines 28-30. The authors say, “The neuromuscular junction (NMJ) is a synapse between the spinal motor neuron and the skeletal muscle fiber, and is a prototypic neuronal synapse of the central nervous system (CNS).” This definition is wrong; the neuromuscular junction is a synapse of the Peripheral Nervous System in which an ectoderm derived element (the axon terminal) contacts with aa mesoderm derived element, the muscle fiber (see Peters, Palay and Webster (1976) The Fine Structure of the Nervous System: The Neurons and Supporting Cells. W.B. Saunders Co., PA, pp. 120-124).

Answer to (i): Thank you for the suggestion. As suggested, we revised our definition on the neuromuscular junction in line 29-30.

Comment (ii) Figure 1. Although mutations eliciting CMS are stated either in red or black, and their description is clear through the text of the paper (lines 166-277). The inclusion of the acronym of each CMS near to the mutation (red or black) in the figure will help to the reader.

Answer to (ii): Thank you for the suggestion. As suggested, we included statements for the red and black mutations in revised Figure 2.

Reviewer 2 Report

This was a well written review that has pulled together a myriad of recent developments in the understanding of the complex processes and interactions involved in NMJ development.

This subject deserves a comprehensive review, and this article meets that need.

Only slight problem. On line 313, abbreviation SMN was not defined. I presume this refers to spinal motor nerve.

Author Response

Response to reviewer 2

Thank you for pointing out our mistake.

Comment: Only slight problem. On line 313, abbreviation SMN was not defined. I presume this refers to spinal motor nerve.

Answer to the comment: We apologize for lack of definition of SMN. As suggested, we indicated that SMN stands for the spinal motor neuron in line 322.

Reviewer 3 Report

Comments on Manuscript ID: ijms-1108373:

“Secreted signaling molecules at the neuromuscular junction in physiology and pathology.”

 In this review, the authors describe the signaling pathways and protein-protein interactions that regulate NMJ formation and maintenance. They also touch upon their role in light of physiology and pathology. The review is well-structured and supported by 2 interesting figures. The main body is divided in a few chapters that describe the different signaling pathways and how the mutations in the involved genes affect the NMJ. This is informative to the readers and covers a broad set of experimental data on NMJ signaling. In general this is a well written review that will interest neuromuscular scientists. The manuscript would benefit from English editing and adjusting the minor comments mentioned below. The quality of writing is significantly better in the second half of the review. It would be good if this is more homogeneous throughout the manuscript.

Minor comments:

  1. Perhaps the abstract could include information on which signaling molecules the review focusses on and what aspects exactly (e.g. mutations, protein-protein interactions…)?
  2. What do the authors mean with the sentence “ is a prototypic neuronal synapse of the central nervous system (CNS)” in the introduction?
  3. In introduction the text “ beneath the synaptic muscle membrane” should be below AChR?
  4. It is unclear what is referred to exactly in the sentence “Rapsyn also binds to other

critical muscle-derived structural molecules, which ensure the efficient generation of an

endplate potential to trigger a muscle action potential.” How does rapsyn ensure this other than clustering AChR? Perhaps this should be specified?

  1. “ More than 30 genes have been identified in CMS patients” mutations in > 30 genes have been identified, not the genes themselves.
  2. The coherence of data in the paragraph starting at line 53 is unclear. It is helpful to summarize the different groups of genes involved in CMS, however the coherence with the second half of this paragraph is unclear. Some info is also double for example how a agrin-MuSk-Lrp4 interaction is composed (line 112 ).
  3. The reviewer really likes figure 2 and wonders whether this can be extended with other panels for example showing the ECM component (chapter 2) and other interactors that are mentioned further down the road. It is unclear why certain signaling pathways are included while others, which were also discussed, are not. This visualization would help the reader to understand the complex mixture of involved proteins and interactions. The level of detail per molecule also differs. For example Lrp4 domains are indicated while MuSK is a stick figure. You could consider to give a bit more detail to all molecules or don’t show it for any. Please also provide a reference in the text to Figure 2 and a bit more detailed captions describing the mechanisms depicted.
  4. Line 133 perhaps explain why these mice die?
  5. The reference in text to fig 1 suggests a visual on how MuSK-Lrp4 and so on induce AChR clustering. However this is figure 2. It is not clear why figure 1 shows certain genes and others not? The authors should reconsider figure 1 and place an appropriate reference in the text to it as it does contain a very nice and useful figure.
  6. MuSK also is engaging in interactions with BMPs and dok7. This should be added to the paragraph covering MuSK
  7. The conclusion section does not match so well with the previous chapters. Perhaps a separate section devoted to therapeutic strategies targeting the mentioned signaling cascades would be better and then keep the conclusions separate. The purpose of this section is now a bit vague.

Author Response

Response to reviewer 3

Thank you for your comprehensive comments. The manuscript had been professionally proofread by a native English speaker.

Minor comments:

  1. Perhaps the abstract could include information on which signaling molecules the review focusses on and what aspects exactly (e.g. mutations, protein-protein interactions…)?

Answer: As suggested, we stated what are introduced in this review article. See lines 17-18.

  1. What do the authors mean with the sentence “is a prototypic neuronal synapse of the central nervous system (CNS)” in the introduction?

Answer: As suggested, we clarified relevant statements in lines 29-31.

  1. In introduction the text “beneath the synaptic muscle membrane” should be below AChR?

Answer: As suggested, we clarified our statement in lines 47-49.

  1. It is unclear what is referred to exactly in the sentence “Rapsyn also binds to other critical muscle-derived structural molecules, which ensure the efficient generation of an endplate potential to trigger a muscle action potential.” How does rapsyn ensure this other than clustering AChR? Perhaps this should be specified?

Answer: As suggested, we explained rapsyn in more detail in lines 47-50.

  1. “More than 30 genes have been identified in CMS patients” mutations in > 30 genes have been identified, not the genes themselves.

Answer: Thank you for pointing this out. As suggested, we revised our statement in line 57

  1. The coherence of data in the paragraph starting at line 53 is unclear. It is helpful to summarize the different groups of genes involved in CMS, however the coherence with the second half of this paragraph is unclear. Some info is also double for example how a agrin-MuSk-Lrp4 interaction is composed (line 112).

Answer: As suggested, we introduced the signaling molecules as the last category of CMS-associated mutations. We also indicated genes included in a category of the signaling molecules in lines 65-66.

  1. The reviewer really likes figure 2 and wonders whether this can be extended with other panels for example showing the ECM component (chapter 2) and other interactors that are mentioned further down the road. It is unclear why certain signaling pathways are included while others, which were also discussed, are not. This visualization would help the reader to understand the complex mixture of involved proteins and interactions. The level of detail per molecule also differs. For example, Lrp4 domains are indicated while MuSK is a stick figure. You could consider to give a bit more detail to all molecules or don’t show it for any. Please also provide a reference in the text to Figure 2 and a bit more detailed captions describing the mechanisms depicted.

Answer: Thank you for your encouraging comments. As suggested, we simplified the shape of Lrp4 molecule. We also rearranged pairs of a secreted molecule and its receptor in the order in text. We also added short statements of secreted molecules with chapter numbers. Fig. 2 was changed to Fig. 1. Please see lines 117-122.

  1. Line 133 perhaps explain why these mice die?

Answer: As suggested, we explained that mice die with respiratory failure at birth. Please see lines 150-151.

  1. The reference in text to fig 1 suggests a visual on how MuSK-Lrp4 and so on induce AChR clustering. However, this is figure 2. It is not clear why figure 1 shows certain genes and others not? The authors should reconsider figure 1 and place an appropriate reference in the text to it as it does contain a very nice and useful figure.

Answer: We apologize for erroneous figure citation. We fixed our error. We also cited new Fig. 1 wherever we think it is appropriate.

  1. MuSK also is engaging in interactions with BMPs and dok7. This should be added to the paragraph covering MuSK

Answer: As suggested, we included BMP4 as one of MuSK-binding partners in lines 258-260. As MuSK binds to so many intracellular molecules including Dok-7, we did not address Dok-7 as a MuSK-binding partner.

  1. The conclusion section does not match so well with the previous chapters. Perhaps a separate section devoted to therapeutic strategies targeting the mentioned signaling cascades would be better and then keep the conclusions separate. The purpose of this section is now a bit vague.

Answer: We noticed that the conclusion section is not mandatory in this journal. We revised the title to “Therapeutic perspectives” in line 419.